# Research Progress in Calcitonin Gene-Related Peptide and Bone Repair

**DOI:** 10.3390/biom13050838

**Published:** 2023-05-15

**Authors:** Qichang Wang, Haotian Qin, Jiapeng Deng, Huihui Xu, Su Liu, Jian Weng, Hui Zeng

**Affiliations:** 1National & Local Joint Engineering Research Center of Orthopaedic Biomaterials, Peking University Shenzhen Hospital, Shenzhen 518036, China; wangqichang2020@email.szu.edu.cn (Q.W.); qht@pku.edu.cn (H.Q.); djpdeng@bjmu.edu.cn (J.D.); xuhuihui@pkuszh.com (H.X.); lius0706@bjmu.edu.cn (S.L.); 2Department of Bone & Joint Surgery, Peking University Shenzhen Hospital, Shenzhen 518036, China; 3School of Clinical Medicine, Department of Medicine, Shenzhen University, Shenzhen 518061, China; 4Shenzhen Key Laboratory of Orthopaedic Diseases and Biomaterials Research, Shenzhen 518036, China

**Keywords:** calcitonin gene-related peptide, osteogenic effect, immunomodulation, bone repair

## Abstract

Calcitonin gene-related peptide (CGRP) has 37 amino acids. Initially, CGRP had vasodilatory and nociceptive effects. As research progressed, evidence revealed that the peripheral nervous system is closely associated with bone metabolism, osteogenesis, and bone remodeling. Thus, CGRP is the bridge between the nervous system and the skeletal muscle system. CGRP can promote osteogenesis, inhibit bone resorption, promote vascular growth, and regulate the immune microenvironment. The G protein-coupled pathway is vital for its effects, while MAPK, Hippo, NF-κB, and other pathways have signal crosstalk, affecting cell proliferation and differentiation. The current review provides a detailed description of the bone repair effects of CGRP, subjected to several therapeutic studies, such as drug injection, gene editing, and novel bone repair materials.

## 1. Overview of CGRP

### 1.1. Structure of CGRP

Calcitonin gene-related peptide (CGRP) is a polypeptide with 37 amino acids [1]. There are two main in vivo forms of CGRP, viz., αCGRP and βCGRP. αCGRP is mainly encoded by the calcitonin gene, also called the CALCA gene. It is synthesized post-transcriptionally through specific splicing, synthesizing two αCGRP and CT different products. βCGRP is encoded by the CALCB gene with >90% structural similarity to αCGRP [2]. αCGRP is mainly distributed in central and peripheral nerve fibers. In contrast, βCGRP is primarily distributed in the gastrointestinal tract, possessing similar functions with slightly higher biological efficacy than αCGRP [3]. The current research focuses on αCGRP, and this review is mainly derived from studies on αCGRP. CGRP has four structural domains: 1. The first has the header seven amino acids at the NH_2_ terminus, with disulfide bonds and a ring structure at residues 2 and 7. They can interact with the transmembrane structural domain of the CLR for receptor activation. The CGRP8-37 antagonist can be generated by removing the first structural domain of CGRP; 2. the second structural domain has 8–18 residues, and their absence causes a 50- to 100-fold decrease in CGRP affinity; 3. the third structural domain includes 19–27 residues, forming a hinge region; 4. the fourth structural domain possesses the remaining 28–37 residues at the COOH end, serving as two receptor-binding epitopes [1,4]. The two structural domains at the NH2 and COOH ends are the most important in the receptor binding process. The COOH end binds to the extracellular structural domain (ECD) of the receptor. In contrast, the disulfide ring at the same NH_2_ end attaches to the transmembrane structural domain (TM) and the extracellular loop (ECL) region. This activates the receptor and transmits signals to the intracellular compartment [5].

### 1.2. Receptors of CGRP

Currently, receptors and related regulatory proteins of CGRP mainly include calcitonin gene-related peptide receptor (CLR) and receptor activity modifying protein 1 (RAMP1). CLR is a seven-transmembrane receptor protein responding to adrenocorticotropin-releasing factor (CRF) and frog skin peptide [6]. However, cell lines constructed using CLR alone had no significant correlation with CGRP stimulation [7]. RAMP1 exists in mammalian cells, which, when co-expressed with CLR, CGRP can bind to CLR to exert biological effects [8]. Follow-up studies revealed that RAMP1 transports CLR to the cell membrane surface. This creates a CLR-RAMP1 dimer for CGRP to bind and function with CLR; thus, RAMP1 is essential for CLR function [9].

In addition, a new CGRP receptor-related protein, receptor component protein (RCP), enhances the CGRP effects [10]. RCP regulates CGRP/CLR-RAMP1 cell signaling [11], and immunoprecipitation indicates mutual binding sites between CLR and RCP. It is unclear whether RCP is necessary for CLR activation by CGRP. Therefore, further experiments must explore the specific function of RCP [12].

### 1.3. CGRP Synthesis, Storage, and Release

CGRP is mainly synthesized in sensory nerve fibers [13], stored, and released as vesicles [14]. The most representative cells are the spinal dorsal root ganglion (DRG) and the trigeminal ganglion. They are the primary sensory neurons and one of the most abundant CGRP sites. Membrane potential alterations lead to the spinal DRG triggering Ca^2+^ influx and activating the Ca^2+^/CaMKII/CREB signaling cascade, synthesizing CGRP. Then, CGRP is transported to peripheral nerve endings and released after membrane depolarization [15]. The trigeminal ganglion is also rich in CGRP; its CGRP-positive neurons account for half [16]. Other sites, such as autonomic ganglia and cerebral vessels, are also rich in CGRP [17,18,19]. The CGRP receptors have a wide range and various biological effects due to wide distribution.

### 1.4. Physiology and Pathophysiology of CGRP

Research has confirmed that the secretion of CGRP has a circadian rhythm in vivo, with a higher amount released into plasma at night [20]. The half-life of CGRP secreted by the body is relatively short, typically less than 10 min. This short half-life is due to the rapid metabolism and clearance of the peptide by various enzymes and organs in the body, such as the liver and kidneys [21]. Under physiological conditions, CGRP primarily exerts its effects locally, particularly in the peripheral nervous and vascular systems, where it mainly mediates vasodilation and pain transmission.

CGRP is a diastolic microvascular agent and mediates vasodilation [22]. CGRP is 10-fold more potent than prostaglandins and 10–100-fold more potent than neuropeptide vasodilators, including ACh and SP. CGRP has selective modulatory effects on cerebral, coronary, and renal vasculature. Moreover, the vasodilatory effect of CGRP is longer lasting than other vasodilators. However, CGRP does not regulate systemic blood pressure in normal individuals [23,24]. It was believed that the powerful vasodilatory effects of CGRP could develop novel antihypertensive drugs. However, in vivo experiments did not correlate CGRP with systemic blood pressure regulation.

CGRP mediates nociceptive transmission CGRP [25], as they act as afferent neurotransmitters after being released by sensory neurons to conduct nociception [26]. CGRP release is associated with somatic, visceral, neuropathic, and inflammatory pain [27]. Research has found that the pain-mediating ability of CGRP is weaker than that of substance P [28]. It is important to note that the efficacy of CGRP observed in different animal models can vary [29]. The role of CGRP in pain has been controversial, with some studies suggesting that it can induce hyperalgesia, or increased sensitivity to painful stimuli [1]. However, there is already much evidence regarding the association between CGRP and migraine [30,31,32]. CGRP is extremely high in the trigeminal ganglion and is selectively released from the trigeminal nervous system during acute migraine. CGRP receptor antagonists and monoclonal antibodies effectively relieve migraine [33]. The pain-relieving effect of CGRP monoclonal antibodies is substantial in treating chronic inflammatory conditions (osteoarthritis, chronic low back pain, migraine or rheumatoid arthritis, etc.) [34]. Developing novel analgesic agents targeting CGRP antagonism is of considerable scientific interest. Table 1 summarizes the most representative CGRP antagonists.

Under pathological conditions, due to the widespread expression of CGRP receptors throughout the body, abnormal CGRP secretion may impact various physiological systems, including the circulatory system, nervous system, skeletal muscle system, and endocrine system. Abnormal CGRP secretion has been associated with an increased risk of neuralgia, ischemic heart disease, atherosclerosis, chronic joint inflammation, obesity, diabetes, and other diseases [1,22,47,48]. Therefore, understanding the role of CGRP in these physiological processes is important for developing effective treatments for these diseases.

## 2. Effects of CGRP on Bone Repair

CGRP is widely distributed in bone tissue and promotes osteogenesis, inhibits osteolysis, induces angiogenesis, and regulates the immune microenvironment (Figure 1).

### 2.1. Distribution of CGRP and Receptors in Bone Tissue

Bone tissue is rich in CGRP, with two types of nerve fibers—substance P (SP)-positive and CGRP-positive nerve fibers. They are mainly present in bone, bone marrow, periosteum, synovium, and adjacent soft tissues. The proportion of CGRP-positive nerve fibers is dominant [49], and CGRP is a critical node that cannot be bypassed and is a bridge between nerve and bone repair. The receptors of CGRP, CLR, and RAMP1 are widely distributed in vivo, and the co-localized expression of CLR and RAMP1 is also crucial for CGRP to exert its effects. In bone tissue, macrophages, osteoblasts, and vascular tissue, CLR and RAMP1 expression is abundant in macrophages, osteoblasts, and endothelial cells [50,51,52,53,54]. This regulates osteogenesis by affecting the immune microenvironment, vasculature, and accompanying nerves. Immune cells promote bone production by secreting certain osteogenic factors [55]. In contrast, blood vessels and nerve fibers are distributed throughout the bone tissue, providing oxygen, nutrients, and supporting cells to the bone tissue. Therefore, they are essential in bone growth, development, and fracture healing [56]. In short, sensory neurons are stimulated to continuously release CGRP into bone tissue to achieve neurological bone regulation.

### 2.2. CGRP Promotes Osteogenesis

Among the neuropeptides identified in bone tissue (CGRP, SP, NE, NPY), CGRP is most strongly associated with bone repair [57]. Mice with a systemic knockout of the CGRP gene have significantly reduced bone mass. In contrast, overexpression of the CGRP gene significantly enhances bone density [58,59]. Therefore, CGRP promotes bone repair accompanied by nerve fiber ingrowth. There were fewer CGRP-positive nerve fibers at inflammatory progression sites (larger defect sites) and more CGRP-positive nerve fibers at repair sites (smaller defects) during knee and ankle osteoarthritis. Thus, inflammation destroying bone may induce the ingrowth of CGRP-positive fibers, producing CGRP for local bone repair [60]. Sequencing results indicate that CGRP promotes extracellular matrix production, which may be an essential pathway for its bone repair effects [54]. Sensory nerves maintain extracellular matrix (ECM) homeostasis through the CGRP/CHSY1 axis, and the knockdown of sensory nerve CGRP induces similar disturbances in ECM metabolism [61].

CGRP can upregulate various osteogenic factors and promote osteoblast anabolism [62,63]. Haitao et al. observed that CGRP promotes elevated levels of cAMP, ATF4, and OCN expression in osteoblasts [64]. ATF4 is an ATF/CREB family member, a cell-specific CREB-related transcriptional essential for osteoblast differentiation and function factor [65]. ATF4 is identified as an osteoblast-specific transcription factor necessary for OCN transcription, an osteoblast-specific marker commonly used to indicate late osteoblast differentiation [66,67]. BMP2 is essential in osteogenesis induction and promotes ECM expression, mainly collagen production and calcium salt formation, using the Smad pathway [68,69,70]. In MG-63 cells, BMP2 can be involved in CGRP-induced osteogenic differentiation, and cAMP/p-CREB upregulation further promotes BMP2 expression [71]. Dental pulp stem cells (DPSCs) possess the qualities of bone marrow mesenchymal stem cells and are the primary source of dentin mineralization. CGRP directly stimulates a 1.8-fold increase in BMP2 mRNA expression in DPSCs, and a 2.8-fold increase in basal levels of cAMP, which promotes dentin formation. Liping et al. observed that CGRP stimulated bone marrow MSC proliferation, upregulated osteoblast gene expression, enhanced alkaline phosphatase activity, and increased calcium nodules in bone marrow mesenchymal stem cells (BMSCs) [72]. Direct action of CGRP on BMSCs elevated the migratory capacity and osteogenic differentiation and inhibited the bone marrow differentiation of MSCs to adipocytes [54,73]. This process involves molecular crosstalk between Wnt/β-catenin and CGRP signaling [74] other than enhanced cAMP response element binding protein 1 in periosteal-derived stem cells (CREB1) and SP7 (osterix, OSX) expression in periosteal-derived stem cells. Thus, osteogenic differentiation of periosteal-derived stem cells is promoted [75]. In vivo studies identified elevated CGRP levels in mice with femoral fractures [76]. Similar upregulatory effects could be seen when serum CGRP expression was examined in femoral neck fracture patients [77]. CGRP was used to transiently act on bone healing tissue and bone formation regulators (IL1b, Ccl7, MMP13, Mrc1), which increased the PPARγ pathway (Adipoq, Fabp4, Scd1, Cfd) members [76]. Thus, local bone defect repairing was induced by releasing CGRP from nerve endings within the periosteum [75]. Electrical stimulation (ES) of the spinal dorsal root ganglion directly enhances the biosynthesis and release of CGRP by activating Ca^2+^ and accelerating femur fracture healing in osteoporotic rats [15]. Therefore, CGRP is essential in the early generation of bone healing tissue by activating osteogenic effect-related pathways. CGRP also promotes the production of osteogenic factors, collagen, and extracellular matrix, facilitating bone repair.

### 2.3. CGRP Inhibits Bone Resorption

CGRP inhibits osteoclasts, increases bone volume, and reduces bone resorption [58,78]. Previous studies have indicated that sympathetic and sensory nerves are cross-linked to the osteoclastic effect. In contrast, CGRP reverses the isoproterenol (Isp)-mediated osteoclastic effect [79] and inhibits the proliferation of granulocyte-macrophage lineage progenitor cells (precursor osteoclast cells) [80]. CGRP increases OCN and OPG in osteoblasts and inhibits RANKL expression [64]. RANKL is a RANK ligand, an activatable NF-κB pathway that promotes the proliferation and effects of osteoclasts. CGRP inhibits osteogenesis by suppressing RANKL expression, inhibiting the NF-κB pathway, and downregulating the expression of osteoclast TRAP and histone K [72]. However, NF-κB also has a vital role in senescence and apoptosis, and attention must be paid to whether CGRP plays a crucial role in their senescence and apoptosis.

### 2.4. CGRP-Induced Angiogenesis

CGRP is vital in blood vessel formation and growth [81] and can indirectly affect bone development and formation [82,83]. Bone development cannot be separated from the expression of CGRP, vascular endothelial growth factor (VEGF), a cluster of differentiation 31 (CD31), and lymphatic vessel endothelial hyaluronan receptor 1 (LYVE1) [84]. CGRP-positive nerve fibers during early embryonic development are present in blood vessels, developing muscles, or developing cartilage bones [85]. Angiogenesis and bone regeneration were observed by local CGRP injection into the defect site in rats [86]. CGRP promotes VEGF expression to enhance endothelial cell proliferation and migration [87,88], a mechanism validated in tumor-associated angiogenesis [89]. Moreover, CGRP promotes endothelial progenitor cell proliferation and restricts apoptosis by inhibiting MAPK signaling [90]. OCN upregulation, ALP gene expression, and increase in mineralized nodules in osteoblast (OB) monoculture and human umbilical vein endothelial cell (HUVEC) OB co-culture systems could be associated with particular cytokine expression regulation in HUVEC by CGRP [91]. Biodegradable biomagnesium implants can accelerate bone repair by upregulating CGRP and promoting angiogenesis [92]. BMSCs overexpressing CGRP can improve bone repair by promoting peripheral angiogenesis in diabetic rats having tibial defects [93]. CGRP release at the bone defect site can contribute to angiogenesis, enhancing repair [94].

### 2.5. CGRP Regulates the Immune Microenvironment

Inflammation occurrence is essential for bone repair [95,96]. Mice with CGRP knockdown exhibit a higher degree of oxidative stress accompanied by reduced nitric oxide synthase (NOS) expression, increased phosphorylated p47 expression, elevated 4 hydroxynonenal (4HNE) levels, and macrophage infiltration [97]. CGRP inhibits LPS-induced TNFα production by macrophages [98] and osteoblasts [99]. Sensory nerve fibers secrete CGRP to inhibit inflammation by suppressing type 1 T helper cytokine production and leukocyte proliferation. CGRP affects the polarization of M0-type to M2-type macrophages, affecting bone tissue remodeling in the later stages of fracture repair [100]. Moreover, CGRP induces bone regeneration and differentiation by elevating M2-type macrophage proportion [101]. Lack of CGRP promotes M1 and inhibits M2 polarization in macrophages. Thus, CGRP knockdown mice hinder osseointegration in bone grafts, and its overexpression improves osseointegration by regulating the macrophage phenotype [102].

However, osteogenic factors (BMP2, BMP6, WNT10b, and OSM) secreted by M2 macrophages were reduced in the early stages and elevated at later stages of CGRP action [103]. Pajarinen et al. indicated that proper inflammation is beneficial in the early stages of bone repair. M1 macrophages may exert their effects in the early and middle stages of osteogenesis. In contrast, M2 macrophages later affect bone matrix mineralization [55], which aligns with the prevailing view of immune regulation of bone repair. Thus, CGRP plays a vital role in regulating the immune microenvironment for bone repair [96,104].

To summarize, CGRP promotes osteogenesis by regulating diversified cells. Table 2 shows a study on CGRP at the cellular level.

## 3. Effect of CGRP on Osteogenic Effect Pathways

### 3.1. Major Pathways of CGRP—G Protein-Coupled Receptor Pathway

The CGRP effector pathway was mainly the G protein-coupled receptor pathway (Figure 2).

When CGRP acts on pancreatic cells, the intracellular cAMP concentration increases persistently [114]. Similar phenomena were observed when CGRP acted on vascular endothelial cells and osteoblasts [107,115]. Subsequent studies observed that Gs protein (Stimulatory G protein) has a high-affinity binding site to CGRP [108,110,116]. Main et al. pointed out that CLR is a G protein-coupled receptor that transmits biological signals downstream by increasing the intracellular cAMP concentration [117].

CLRs belong to the class B “secretin” family of G protein-coupled receptors (GPCRs) [1] with significant biological effects of CGRP. Two major classical GPCR signaling pathways are 1. Gαs-CLR/cAMP/PKA pathway: CLR first binds to RAMP1, transports CLR to the cell surface, and couples with Gαs-type G proteins to activate the cell surface enzyme adenylate cyclase. This converts adenosine triphosphate (ATP) to cyclic adenosine monophosphate (cAMP), followed by protein kinase A (PKA) activation. PKA continues to activate cyclic AMP effector element binding protein (CREB) into the nucleus to produce active phosphorylated cyclic AMP effector element binding protein (p-CREB). This promotes transcription of osteogenic-related factors, including BMP2, RUNX2, and SP7, and exerts osteoinductive effects; 2. Gαq/11-CLR/IP3/Ca^2+^ and Gαq/11-CLR/DAG/PKC pathways: the increased phospholipase C β1 (PLC_β1_) activity of Gαq/11 cleaves phosphatidylinositol-4,5-bisphosphate (PIP_2_) to inositol-1,4,5-trisphosphate (IP_3_) and diacylglycerol (DAG). IP_3_ acts on the IP_3_ receptor on the endoplasmic reticulum membrane, leading to Ca^2+^ efflux into the cytoplasm. Moreover, diacylglycerol promotes protein kinase C (PKC) activity [118,119] and downward activation of osteogenic protein expression. However, intracellular cAMP concentration was much higher than Ca^2+^ after CGRP treatment. Therefore, CGRP may favor the Gαs-CLR/cAMP/PKA pathway with a downward signal for biological effects [120].

### 3.2. Other Signal Pathway Crosstalk

The non-G protein-coupled pathway for the osteogenic effect of CGRP is less studied. The classical Wnt signaling pathway is an important regulator involved in bone metabolism [121]. CGRP can inhibit the apoptosis of osteoblasts by regulating the classical Wnt signaling pathway [109]. Moreover, CGRP participation can activate the Wnt pathway to promote BMSCs to osteoblast lineage differentiation [74]. The osteogenic differentiation ability of BMSCs is modulated by affecting the Hippo/Yap pathway [106]. CGRP also affects the Hippo/Yap pathway to control the osteogenic-inducing factor secretion by M2-type macrophages [103]. CGRP promotes BMSC differentiation toward endothelial cells and promotes vascular ingrowth using the PI3K-AKT pathway [86]. CGRP reduces RANKL, increases OPG levels, and inhibits bone resorption by inhibiting the NF-κB pathway [111]. However, the above pathways are not directly activated by CGRP, which possesses specific binding receptor CLR, and receptor-regulated protein RAMP1. Therefore, activating the above pathways is more likely to signal crosstalk generated by activating G protein-coupled pathways. More future studies are needed to determine the effects of CGRP pathway activation.

## 4. CGRP in Orthopedic Treatment

### 4.1. Prospects for the Application of CGRP as a Drug

Scientists accelerate the clinical conversion of CGRP by performing animal experiments. Table 3 shows a study on CGRP at the animal level. Scientists consider that CGRP has good application prospects as a drug.

CGRP drug has not been used in clinical practice despite having anti-osteoporotic and osteoinductive effects [63]. Regular intravenous supplementation of CGRP in aged osteoporotic mice significantly increased osteogenesis and decreased bone marrow fat accumulation [54,105]. CGRP carriage as a drug into calcium phosphate bone cement (Sr-CPC) significantly enhanced cell proliferation and elevated ALP secretion in BMSCs [124]. Adipose mesenchymal stem cells (ADSCs) overexpressing CGRP promote the expression of collagen type I (COL1) and bone bridging protein (OPN) with a stronger potential to differentiate into osteoblasts in vitro. Moreover, transplantation of ADSCs overexpressing CGRP into bone defect sites can cause osteopathic effects [113]. BMSCs overexpressing CGRP also possessed better osteogenic effects. Thus, BMSCs carried into collagen scaffolds were more capable of repairing cranial defects in rats [125]. However, direct CGRP trigger supplementation enhances local pain, suggesting a debatable medicinal value of CGRP.

### 4.2. Biomaterials Can Promote Bone Repair by Modulating CGRP

Although the CGRP drug has not been intensively studied, promoting CGRP expression in vivo for osteogenesis is a potent research topic. Previous studies demonstrated that CGRP has a vital role in bone repair. CGRP-positive nerve fibers are widely distributed in the periosteum and bone marrow [126]. Promoting CGRP release and elevating its local concentration is a more reliable fracture treatment to accelerate bone healing. CGRP is mainly found in vesicles in vivo. Researchers have extracted extracellular vesicles (EV) from adipose mesenchymal stem cells and carried CGRP-rich EV in polylactic acid-hydroxyacetic acid copolymer (PLGA) to enhance the growth of alveolar bone and improve bone repair [128]. Magnesium (Mg) could promote CGRP release and accelerate the bone repair process [75,94]. Mg and its alloys have good biodegradability and biocompatibility and can become bone-replacement materials [129]. Larger healing bone tissues with significantly increased mechanical strength were observed when Mg rods were implanted at the fracture site in osteoporotic rats. Mg promotes osteogenic effects by releasing CGRP from the dorsal root ganglion (DRG) and periosteal nerve fibers and activating cAMP/CREB signaling, wherein periosteal-derived stem cells were identified as CGRP targets [75]. A follow-up study observed that implanting Mg nails into the bone defect site and distraction osteogenesis elevated CGRP concentration in the new bone tissue. Furthermore, this accelerated vascular growth into the bone defect site and bone repair through the CGRP/FAK/VEGF signaling axis [94].

### 4.3. Modulation of CGRP for Bone Repair by Electrical Stimulation

Electrical stimulation promotes the long entry of CGRP-positive nerve fibers inside the damaged area [130]. Researchers have promoted the biosynthesis and release of CGRP directly during discharge by placing electrodes in the dorsal root ganglion area of the lumbar spine. This enhances osteoporotic fracture healing within the rat femur [15].

## 5. Other Insights on CGRP and Bone Repair Studies

Therefore, CGRP is an essential link in bone repair, but various voices have emerged. Hoff et al. observed that the offspring of mice knocked out the CALCA gene (equivalent to knocking out both CT and CGRP) had no individual developmental defects. However, there was an increase in bone mass, with a significant increase in bone trabeculae volume and a 1.5- to 2-fold enhancement in bone formation at 1 and 3 months of age [122]. Wear particles produced by joint prostheses induce bone resorption. CALCA gene-deficient wear particle-stimulated aged mice indicated a significant increase in OPG and OCN and a significant decrease in RANKL compared to wild-type mice. Thus, CGRP downregulation in aged mice could enhance osteoprotective effects [123].

BMPs may increase sensory neurogenic CGRP expression [131], and BMP2 is essential for bone formation. However, the relationship between BMP2 and CGRP is quite specific, with substance P synergizing with BMP2 to elevate the osteogenic differentiation of MC3T3-E1 and C2C12. In contrast, the concomitant addition of CGRP reverses the osteoinductive effect, significantly inhibiting osteogenesis [132]. Although the osteoinductive effect of CGRP is well-established, the mechanistic studies are underdeveloped due to harsh in vitro culture conditions of neuronal cells. Moreover, the progress of the in vitro studies lags behind the in vivo studies, and the osteoinductive effect of CGRP could result from the characteristic conditions. These findings are inconsistent with previous studies, which should be addressed in future studies.

## 6. Conclusions and Perspectives

CGRP is an important neuropeptide, and previous studies have found that it has important effects such as vasodilation and pain conduction. With the deepening of research, the important role of CGRP in promoting bone repair has also been revealed [133]. All tissues and organs of the body do not work independently. The nervous system promotes bone repair by secreting CGRP and acting on osteoblasts, osteoclast, vascular endothelial cells, and macrophages. CGRP may be an important bridge connecting the nervous system, circulatory system, immune system, and skeletal muscle system. In vitro studies found that CGRP exerts its biological effect by activating the CLR-RAMP1 dimer on the cell surface. CLR belongs to the G protein-coupled receptor, and RAMP1 is the regulatory protein of CLR. Together, CGRP can promote CREB entering the nucleus and phosphorylation activation, enhance mRNA transcription, and promote the expression of osteogenic factors such as BMP2 and SP7 to promote bone formation by increasing the concentration of intracellular cAMP. In vivo studies have found that CGRP can promote bone formation, promote vascular growth, inhibit osteoclasis, and regulate the immune microenvironment to promote bone repair. When the body is injured, the CGRP at the fracture site can quickly reach the peak, promote the formation of a larger callus locally, and directly enhance the expression of osteogenic factors such as BMP2 and SP7. As time goes on, the CGRP effect gradually decreases, and bone formation slows down. At this time, the activity of osteoclast increases, and then the ability of bone remodeling increases. In terms of immune regulation, early pro-inflammatory responses are crucial for bone repair. CGRP can inhibit the function of M2 macrophages in the early stage and promote the expression of M2 macrophages and osteogenic-related factors in the later stage. In addition, CGRP can also promote the generation of vascular lymphatic vessels and promote the occurrence of distraction osteogenesis.

Developing peptide drugs that promote bone formation and inhibit osteoporosis based on CGRP has good prospects. Peptides as therapeutic agents have advantages such as high efficiency, high selectivity, and low toxicity. However, many peptides have disadvantages such as poor oral bioavailability, strong hydrophobicity, and fast metabolism in vivo [134]. Natural peptides are usually not suitable for direct clinical use as therapeutic agents. Overcoming the shortcomings of natural peptides through reasonable design is the future direction of peptide therapy [135]. Previous studies have found that CGRP is closely related to various types of pain, and CGRP receptors are widely expressed. If CGRP drugs are used throughout the body, there may be many side effects. Currently, there are no literature reports on the application of CGRP drugs in bone repair, and avoiding CGRP-mediated pain and vascular relaxation to highlight their bone repair effect is a difficulty in related research. More clever structural design of CGRP drugs can serve as an important breakthrough direction. Although CGRP has been limited in drug development, local drug delivery systems and in vivo CGRP activation can still serve as the development direction of CGRP therapy, including stent-carrying CGRP, electrical stimulation of nerve cell secretion of CGRP, and magnesium ions promoting CGRP secretion and subsequently bone repair, which have been validated [15,75,94,128]. CGRP promoting bone repair is the focus of future research. Our team has currently conducted research on magnesium ion regulation of CGRP promoting bone repair, and on the enhancement of osteogenic effect by overexpressing CGRP in BMSCs. This may provide new ideas for the clinical transformation of materials science and stem cell therapy, and also open up a new path for the therapeutic efficacy of bone repair. Therefore, CGRP is an essential link in communicating nerve-bone repair, with promising applications in bone repair.

## Figures and Tables

**Figure 1 biomolecules-13-00838-f001:**
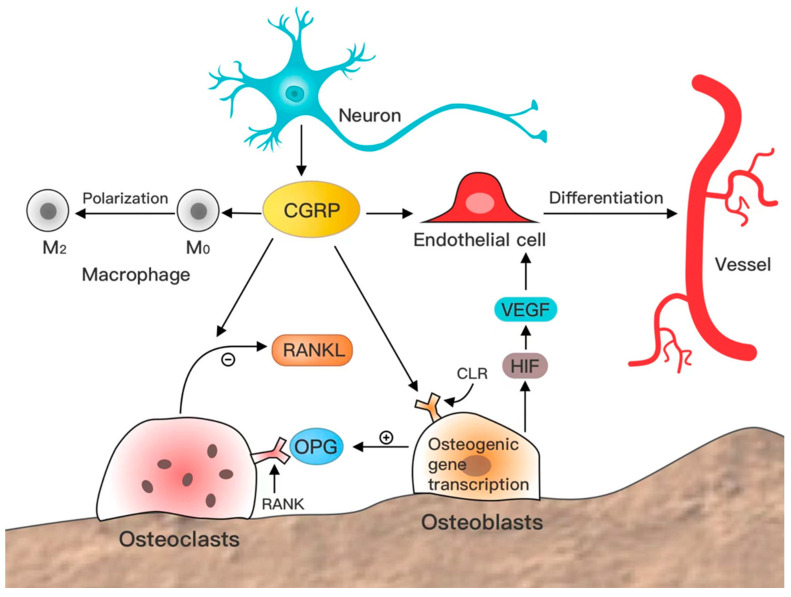
CGRP regulates various cells to promote bone repair. CGRP is secreted by sensory nerve cells, and its effects on bone tissue are: 1. CGRP can bind to the CLR receptor on the cell surface to directly enhance the osteoblast and osteogenic gene expression; 2. CGRP can inhibit the RANKL/RANK/NF-κB pathway by increasing OPG level, inhibiting osteoclast proliferation, and reducing the osteoclast effect; 3. CGRP can enhance the angiogenesis of vascular endothelial cells by promoting the expression of VEGF and HIF factors, induce angiogenesis, and provide nutritional support for bone repair; 4. CGRP can enhance the transformation of M0 into M2 macrophages and regulate the immune microenvironment of bone repair. CGRP is a calcitonin gene-related peptide, CLR is a calcein gene-related peptide receptor, RANK is the receptor activator of nuclear factor kappa B, RANKL is the receptor activator of nuclear factor kappa B ligand, OPG is osteoprotegerin, HIF is a hypoxia-inducible factor, and VEGF is a vascular endothelial growth factor.

**Figure 2 biomolecules-13-00838-f002:**
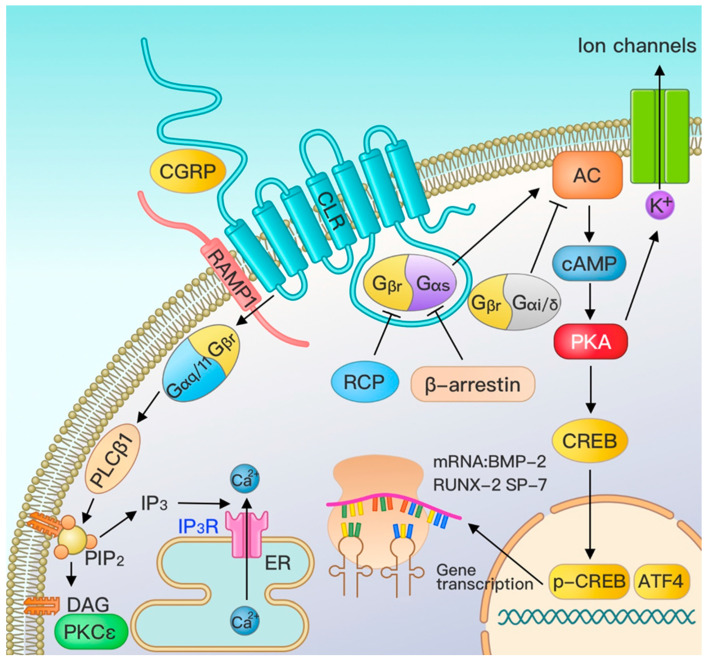
CGRP cellular pathway. CGRP transmits signals to the cell using the G protein-coupled receptor-CLR-RAMP1 dimer: 1. cAMP signaling pathway: activates the G protein and AC to increase the intracellular second messenger. cAMP concentration activates PKA and CREB and promotes mRNA transcription in the nucleus. 2. Phosphatidylinositol signaling pathway: by activating PLCβ1, hydrolyzing PIP2 to generate second messengers, IP3 and DAG. IP3 is gated with the IP3 ligand on the endoplasmic reticulum. Calcium channel binding elevates the intracellular Ca^2+^ concentration and activates various calcium ion-dependent proteins. DAG rivets on the cell membrane to activate PKC bound to the cell membrane. Moreover, intracellular free PKC can phosphorylate the serine/threonine residues of proteins and have a biological effect. AC is adenylate cyclase. cAMP is cyclic adenosine monophosphate, PKA is protein kinase A, CREB is cyclic adenosine monophosphate response element binding protein, PLCβ1 is the phospholipase C, β1, IP3 is inositol 1,4,5-triphosphate, DAG is the diacylglycerol, PIP2 is the phosphatidylinositol-4,5-bisphosphate, PKC is a protein kinase C, CGRP is a calcitonin gene-related peptide, CLR is a calcitonin gene-related Peptide receptor, RAMP1 is the receptor activity modifier protein 1, RCP is the receptor component protein, and ER is the endoplasmic reticulum.

**Table 1 biomolecules-13-00838-t001:** CGRP antagonists.

Target Spot	Name	Type	Current Status
CGRPR	Olcegepant [35]	Nonpeptide	Phase II Clinical Trial
CGRPR	Telcegepant [36]	Nonpeptide	Phase III Clinical Trial
CGRPR	MK-3207 [37]	Nonpeptide	Phase II Clinical Trial
CGRPR	Zavegapant [38]	Nonpeptide	Phase II/III Clinical Trial
CGRPR	BI 44,370 [39]	Nonpeptide	Phase II Clinical Trial
CGRPR	Erenumab [40]	Monoclonal antibody	FDA approved in 2018
CGRPR	Rimegepant [41]	Nonpeptide	FDA approved in 2020
CGRPR	Ubrogepant [42]	Nonpeptide	FDA approved in 2020
CGRPR	Atogepant [43]	Nonpeptide	FDA approved in 2022
CGRP	Galcanezumab [44]	Monoclonal antibody	FDA approved in 2018
CGRP	Fremanezumab [45]	Monoclonal antibody	FDA approved in 2018
CGRP	Eptinezumab [46]	Monoclonal antibody	FDA approved in 2020

**Table 2 biomolecules-13-00838-t002:** Study on CGRP at the cellular level.

Cell Category	Author	Year	Signal Pathway	Experimental Results
BMSCs	Liping Wang [72]	2010	RANKL/NF-κB	Enhance the osteogenic differentiation ability and inhibit osteoclast formation
	Wei Liang [63]	2015		ALP, collagen I, BMP2, osteonectin, and RUNX2 are upregulated, leading to increased proliferation and osteogenic differentiation
	Ri Zhou [74]	2016	Wnt/β-catenin	Enhance the osteogenic differentiation ability
	Jie Chen [105]	2017		The proliferation and osteogenic differentiation abilities are enhanced, and the adipogenic differentiation is inhibited in a dose-dependent manner
	Sen Jia [73]	2019		The expression of ALP and RUNX2 is upregulated, and the proliferation and migration abilities are enhanced
	Qin Zhang [106]	2019	Hippo/Yap	Upregulate ALP, RUNX2, and OPN
	Yanjun Guo [93]	2020		Enhanced expression of VEGF, ALP, and OPN and reduced production of reactive oxygen species (ROS) induced by high glucose
	Hang Li [54]	2021		The osteogenic differentiation ability is enhanced, and the adipogenic differentiation ability is weakened
Osteoblast	Michelangeli [107]	1989	cAMP	Upregulate cAMP
	Vignery [62]	1996	cAMP	Insulin-like growth factor expression is upregulated, and the osteogenic differentiation ability is elevated
	Millet [99]	1997	cAMP	Inhibit the TNF-α production stimulated by lipopolysaccharide and IL-1, but promote IL6 expression
	Villa [108]	2000	cAMP	Enhance the proliferation ability
	Mrak [109]	2010	Wnt/β-catenin	Inhibit apoptosis
	Yang Bo [91]	2013		The expression of OC, ALP, and collagen is upregulated, and the osteogenic ability differentiation is increased
	Haitao He [64]	2016	RANKL/NF-κB	ATF4 and OC expression is upregulated, and the osteogenic differentiation ability is enhanced
Macrophage	Owan [110]	1994	cAMP	Inhibit osteoclast formation
	Feng [98]	1997	cAMP	Inhibit LPS-induced TNF-α upregulation
	Yeong-Min [111]	2014	RANKL/NF-κB	Upregulate OPG expression and inhibit bone resorption
	Qin Zhang [103]	2021	Hippo/Yap	Inhibit and promote the osteogenic factor secretion in M2 macrophages in the early and late stages, respectively
Osteoclast	Akopian [80]	2000	cAMP	Inhibit osteoclast formation in a dose-dependent manner
	Ishizuka [79]	2005	RANKL/NF-κB	Inhibit osteoclast formation
EPCs	Haegerstrand [50]	1990	cAMP	Enhance the proliferation ability
	Shuai Zheng [87]	2010	cAMP	Activate AMPK-eNOS and enhance angiogenesis
	Yang Bo [91]	2013		Upregulate OC, ALP, and COL expression in vascular endothelial cells and osteoblast co-culture system
	Jianqun Wu [90]	2018	MAPK	Enhance the proliferation ability and inhibit apoptosis
	Jie Mi [86]	2021	PI3K/AKT	The proliferation ability and angiogenesis are enhanced. The osteogenic differentiation ability of BMSCs is increased
	Ye Li [94]	2021	FAK/VEGF	Enhanced migration ability, promote FAK phosphorylation of and upregulate VEGF expression
PDSCs	Yifeng Zhang [75]	2016	cAMP	SP7 and ALP expression is upregulated, and proliferation and osteogenic differentiation ability are enhanced
DPSCs	Calland [112]	1997	cAMP	BMP2 expression is upregulated, and the osteogenic differentiation ability is elevated
MG63	Gang Tian [71]	2013	cAMP	BMP2 expression is upregulated, and the osteogenic differentiation ability is enhanced
ADSCs	Zhong Fang [113]	2013		ALP expression in the overexpressed CGRP group is upregulated, and the cell proliferation and osteogenic differentiation ability are enhanced
DRG	Jie Mi [15]	2021	cAMP	Electrical stimulation promotes the synthesis and release of CGRP in DRG and enhances the H-type blood vessel formation and osteoporotic fracture healing

**Table 3 biomolecules-13-00838-t003:** Study on CGRP at the animal level.

Model	Author	Year	Modeling Method	Phenotype
Gene editing	Ballica [58]	1999	Construction of mouse model of CGRP overexpression	Inhibit osteoclasts, stimulate insulin-like growth factor, and inhibit the tumor necrosis factor-α production
	Hoff [122]	2002	Construction of CGRP knockout mouse model	The bone mass of gene-knockout mice was maintained after ovariectomy, and that of gene-knockout wild-type mice decreased within two months.
	Schinke [59]	2004	Construction of CALCA and CGRP knockout mouse model	CALCA knockout mice revealed a high bone mass, while CGRP knockout mice showed a low bone mass.
	Toda [81]	2008	Construction of CGRP knockout mouse model	The expression of vascular endothelial growth factor within wound granulation tissue of CGRP knockout mice decreased. Angiogenesis and wound closure was significantly inhibited.
	Lei Yang [97]	2013	Construction of CGRP knockout mouse model	CGRP can inhibit oxidative stress and the proliferation of vascular smooth muscle cells induced by vascular injury.
	Kauther [123]	2013	Construction of CGRP knockout mouse model	In mice, OPG and OCN increased significantly, osteoclasts elevated, and RANKL decreased significantly.
	Takahashi [78]	2016	Construction of TRPV1 knockout mouse model	TRPV1 affects osteoclast formation by CGRP regulation
	Niedermair [100]	2020	Construction of CGRP knockout mouse model	CGRP relieved pain and promoted the polarization of M2 macrophages but did not affect bone maturation.
	Appelt [76]	2020	Construction of CGRP knockout mouse model	The number of bone-forming osteoblasts in CGRP-deficient mice decreased significantly, and bone healing was poor.
Bone graft	Zhong Fang [113]	2013	Implantation of CGRP overexpressed ADSCs/β-TCP Bracket	ADSCs overexpressing the CGRP/β-TCP stent promotes bone repair
	Wei Liang [124]	2016	Implantation of CPC containing CGRP and Sr	CPC containing CGRP and Sr promotes bone repair among osteoporotic rats
	Xijiao Yu [125]	2019	Implantation of CGRP overexpressed collagen scaffolds in BMSCs	BMScs collagen scaffold overexpressing CGRP leads to skull repair in rats
	Sen Jia [73]	2019	Construction of a rat model of distraction osteogenesis	CGRP enhances new bone formation by elevating the migration and differentiation of bone marrow stromal cells.
	Ye Li [94]	2021	Construction of a rat model of distraction osteogenesis	Magnesium-containing intramedullary nail promotes bone defect repair in rats by the upregulating CGRP/FAK/VEGF pathway.
	Wangyong Zhu [92]	2021	Construction of a rat model of drug-related osteonecrosis	Magnesium grafts promote angiogenesis and bone repair by regulating VEGF and CGRP, thereby alleviating drug-related osteonecrosis.
Injection of drugs	Mapp [88]	2012	Normal knee joint	Promote angiogenesis
	Yanjun Guo [93]	2020	Construction of diabetic rat model	CGRP injection overexpressed BMSCs can elevate ALP activity and promote mRNA and protein expression of VEGF, ALP, and OPN
	Hang Li	2021	Construction of senile mouse model and osteoporotic mouse model	Promote bone formation in aged mice, decrease fat accumulation, and delay osteoporosis occurrence in mice.
Other	Aoki [82]	1994	Tibial bone defect model	Local blood flow and callus formation increased.
	Irie [126]	2002	Normal bone tissue	There are abundant CGRP-positive nerve fibers around the bone tissue.
	Maeda [84]	2017	Normal fetal rat bone tissue	CGRP is essential in osteogenesis and angiogenesis in bone development.
	Feng Gao [127]	2018	Construction of obese mouse model	In obese mice, CGRP, FGF, and TGF-β levels decreased, while TNF-α levels increased, and bone repair was delayed.

## Data Availability

Not applicable.

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
