# Peer review of "Research Progress in Calcitonin Gene-Related Peptide and Bone Repair"

_biomolecules, 2023, doi:10.3390/biom13050838_

Round 1

Reviewer 1 Report

The present review gives an overview of calcitonin, summarizing the latest research on calcitonin. However, it would be good if the authors could give us an overview of the behavior of this peptide at the physiological level and in pathological conditions. Also, what is its role in pain control?. This information could further clarify its potential use in different fields.

Minor editing changes are needed

Reviewer 2 Report

Reviewer comments- biomolecules-2360247

This manuscript describes “Research progress in calcitonin gene-related peptide and bone repair”. This is an interesting topic, and a well written review article on various aspects of calcitonin gene-related peptide and bone repair. However, there are some major and minor issues in the current manuscript. After addressing following concerns, this article can be considered for publication.

Major and Minor concerns:

·      In introduction, as this article related to peptides and its utility for bone repair, so it would be better if authors can add some discussion about peptide drug discovery. As there is already number of peptide-based drugs already in clinic so some discussion would increase readers interest to this peptide work. Please see reference: https://doi.org/10.1016/j.drudis.2022.103464  (Drug Discovery Today (2022): 103464.) andhttps://www.sciencedirect.com/science/article/pii/S1359644614003997 (Drug discovery today 20, no. 1 (2015): 122-128.); and these can be cited. 

·      As Currently, FDA-approved drugs acting on the CGRP, or its receptor, It would be better if author can also discuss about approved drug on CGRP, or its receptor in introduction.   

·      For “Figure 1 and Figure 2”, can author clarify whether they drew these figures or copied from some other article? 

·      Conclusion and perspectives need serious improvements. As this is not conclusion and there is no perspective. 

·      All reference should in uniform pattern as multiple patterns used. 

NA
